# Impact of Environmental Conditions on the Protein Content of *Staphylococcus aureus* and Its Derived Extracellular Vesicles

**DOI:** 10.3390/microorganisms10091808

**Published:** 2022-09-09

**Authors:** Brenda Silva Rosa da Luz, Vinícius de Rezende Rodovalho, Aurélie Nicolas, Svetlana Chabelskaya, Julien Jardin, Valérie Briard-Bion, Yves Le Loir, Vasco Ariston de Carvalho Azevedo, Éric Guédon

**Affiliations:** 1INRAE, Institut Agro, STLO, F-35000 Rennes, France; 2Laboratory of Cellular and Molecular Genetics, Institute of Biological Sciences, Federal University of Minas Gerais, Belo Horizonte 31270-901, Brazil; 3BRM (Bacterial Regulatory RNAs and Medicine) UMR_S 1230, Inserm 1230, University of Rennes 1, 35000 Rennes, France

**Keywords:** membrane vesicle, exosome, virulence factors, vancomycin, proteomics, communication, adaptation

## Abstract

*Staphylococcus aureus*, a major opportunistic pathogen in humans, produces extracellular vesicles (EVs) that are involved in cellular communication, the delivery of virulence factors, and modulation of the host immune system response. However, to date, the impact of culture conditions on the physicochemical and functional properties of *S. aureus* EVs is still largely unexplored. Here, we use a proteomic approach to provide a complete protein characterization of *S. aureus* HG003, a NCTC8325 derivative strain and its derived EVs under four growth conditions: early- and late-stationary growth phases, and in the absence and presence of a sub-inhibitory concentration of vancomycin. The HG003 EV protein composition in terms of subcellular localization, COG and KEGG categories, as well as their relative abundance are modulated by the environment and differs from that of whole-cell (WC). Moreover, the environmental conditions that were tested had a more pronounced impact on the EV protein composition when compared to the WC, supporting the existence of mechanisms for the selective packing of EV cargo. This study provides the first general picture of the impact of different growth conditions in the proteome of *S. aureus* EVs and its producing-cells and paves the way for future studies to understand better *S. aureus* EV production, composition, and roles.

## 1. Introduction

*Staphylococcus aureus* is a Gram-positive bacterium acting both as a commensal and as an important opportunist pathogen. This versatile bacterium causes a broad spectrum of diseases ranging from minor to severe infections in several mammalian species, particularly humans and cattle [1,2,3]. The ability of *S. aureus* to colonize a diversity of hosts and niches in its hosts as well as to cause a vast array of diseases is reflected by the expression of numerous virulence factors, such as adhesins, toxins, and other elements that are capable of promoting host cell invasion, evasion, and pathogenesis [4,5,6,7]. Several virulence factors are secreted in the extracellular milieu, while others are surface-exposed in the *S. aureus* cell wall. Therefore, extracellular vesicles (EVs) are vehicles for transporting and delivering these elements. Indeed, this long-known phenomenon was first described in Gram-positive bacteria in 2009, when Lee et al. reported EV release by *S. aureus* [8].

EVs can be described as biological bubbles for intercellular communication that are released by cells in all domains of life [9,10,11]. Even though they were first judged as simple cell trash, they are currently recognized as essential elements in cell-to-cell interactions [12,13,14,15]. EVs are spheres that are formed of a lipid bilayer ranging from 20 to 300 nm that wraps and protects biological compounds such as nucleic acids, metabolites, and proteins from the action of external elements [9,11]. These bioactive molecules gain a ticket to a bubble ride in EVs, which can travel through close or distant sites from where they were produced to exert their functions [16,17]. Bacterial EVs play roles in bacterial physiology, resistance, competition, and host-pathogen interactions [13,18,19,20,21,22].

The study of EVs is an emerging field in medical and veterinary medicine of *S. aureus* since several reports have highlighted their contribution to cargo delivery, signaling, and cell–cell communication, contributing to physiological and pathological processes [23]. Indeed, *S. aureus* EVs carry several virulence factors (e.g., β-lactamases, toxins, adhesins) that exert essential functions, including the transfer of antibiotic resistance to susceptible bacteria, host cell death, immunomodulation, and exacerbation of inflammatory processes [8,24,25,26,27,28,29,30,31,32,33,34,35,36,37,38].

During infection, *S. aureus* undergoes several physiological states (e.g., growth phases) and stresses (e.g., antibiotic treatment) that affect bacterial metabolism, gene expression, and consequently, EV production and composition. Indeed, significant differences in particle size and/or concentration were reported in response to different conditions, such as growth media [39], growth phases [40], and growth temperatures [27,40], as well as after exposure to environmental stresses, such as iron-depletion, ethanol, oxidative, osmotic stresses [27], and antibiotics [41,42,43]. Concerning EV composition, a recent study by Briaud et al. demonstrated that temperature variations affect the EV protein and RNA cargo quantitatively and qualitatively, which is also reflected in different EV cytolytic activities towards host cells [40]. Culture media, growth phase, and antibiotic treatment were also shown to alter *S. aureus* EV content [39,43,44,45]. However, studies comparing the impact of different environmental conditions on both *S. aureus* producing cells and its derived EVs are still lacking.

In this context, this work aims to explore the protein content of *S. aureus* strain HG003 and its derived EVs under the influence of four conditions: early- and late-stationary growth phases, and in the absence and presence of a sub-inhibitory concentration of vancomycin. The protein cargo was compared between producing cells and EVs, shedding light on how the environment influences *S. aureus* EV cargo packing.

## 2. Materials and Methods

### 2.1. Bacterial Strains and Cultures

The *S. aureus* strain that was used in this work was the model strain HG003 [46], a derivative of NCTC8325 that was isolated in 1960 from a sepsis patient. HG003 contains functional *rsbU* (coding for an activator of the sigma factor B) and *tcaR* (coding for an activator of protein A transcription) genes, two global regulators missing in the NCTC8325 parent strain. The HG003 genome is well documented [47] and this strain is widely used as a reference to investigate staphylococcal regulation and virulence [48]. Here, *S. aureus* HG003 was grown in four experimental conditions: early- and late-stationary growth phases (6 and 12 h, respectively), an in the absence (V−) and presence (V+) of a sub-inhibitory concentration of vancomycin (0.5 μg/mL) that do not affect the HG003 growth [44]. Note that the minimum inhibitory concentration (MIC) of vancomycin for HG003 strain is 0.5 mg/mL [49]. Pre-inoculums and cultures were grown overnight in BHI broth at 37 °C under 150 rpm/min agitation. The bacterial growth was determined by the measurement of the optical density at 600 nm.

### 2.2. EVs Isolation and Purification

EVs were isolated and purified as previously described [44]. Briefly, 1 L of bacterial cell culture was centrifuged at 6000× *g* for 15 min and filtered through 0.22 μm Nalgene top filters (Thermo Scientific, Waltham, MA, United States). Then, the culture supernatant fraction was concentrated around 100-fold using the Amicon ultrafiltration systems (Millipore, Burlington, MA, United States) with a 100 kDa filter and ultra-centrifuged for 120 min at 150,000× *g* to eliminate the soluble proteins. Next, the suspended pellet was applied to a discontinuous sucrose gradient (8–68%) and ultra-centrifuged at 100,000× *g* for 150 min. The fractions containing EVs were recovered and washed in TBS (150 mM NaCl; 50 mM Tris-Cl, pH 7.5) for final ultra-centrifugation at 150,000× *g* (120 min). At last, the EVs were suspended in cold TBS and kept at −80 °C until use. The quality of the EV samples (i.e., homogeneity, integrity, and reproducibility) was assessed by transmission electron microscopy (TEM), Nano Tracking Analysis (NTA) and SDS-PAGE.

### 2.3. Physical Characterization of EVs

The EVs’ size and concentration were determined by NTA (NanoSight NS300, Malvern Panalytical, Worcestershire, United Kingdom) as previously described [44]. The EVs were negatively stained with 2% uranyl acetate and analyzed with JEOL 1400 Electron Microscope (Jeol, Tokyo, Japan) as previously described [44].

### 2.4. Proteome of S. aureus HG003 and Its Derived Vesicles

#### 2.4.1. Protein Extraction and Visualization

The same bacterial cultures were used for EV and whole-cell (WC) protein extraction. Aliquots of 10 mL of bacterial cells were centrifuged and rinsed twice with 10 mL of TBS buffer and suspended in 10 mL of lysis solution (1.52 g Tris-HCL, 0.03 g SDS, 0.3 g DTT, pH 7.5, 10 mL H_2_O). For WC, mechanic lysis was performed with Precellys (6500 rpm, 2 × 30 s), and protein extracts were recovered after 30 min centrifugation at 14,000× *g* rpm. Since EVs lack cell walls, lysis is achieved with a lysis solution. Using silver staining, the bacterial WC and EV protein profiles were visualized by SDS-PAGE [50,51].

#### 2.4.2. Protein Identification and Quantification

For NanoLC-ESI-MS/MS analysis, three independent biological replicates of WC and EV protein extracts (approximately 30 µg per sample) were resolved using 12% SDS-PAGE and Biosafe Blue Coomassie coloration. Then, gel sections were cleansed with acetonitrile and ammonium bicarbonate solutions and dried under a SpeedVac concentrator (SVC100H, Savant Instruments Inc, New York, NY, United States). The samples were submitted to overnight in-gel trypsinization at 37 °C [52,53]. Peptide separation and detection by mass spectrometry were performed according to Tarnaud et al. [54]. The X!TandemPipeline software [55] was used to identify peptides (maximum e-value of 0.05) from the MS/MS spectra. The peptides were searched against the two genomes sequence of *S. aureus* NCTC8325 and HG003 (GenBank accessions no. NC_007795.1 and GCA_000736455.1). The database search parameters were specified as follows: trypsin cleavage was used and the peptide mass tolerance was set at 10 ppm for MS and 0.05 Da for MS/MS. Methionine oxidation and serine or threonine phosphorylation were selected as a variable modifications. For each peptide that was identified, a minimum e-value that was lower than 0.05 was considered to be a prerequisite for validation. A minimum of two peptides per protein was imposed, resulting in a protein false discovery rate (FDR) < 0.5% for peptide and protein identifications. Note that a protein was considered present in a given condition when it was detected in at least two out of three biological replicates. Each peptide that was identified by tandem mass spectrometry was quantified using the free MassChroQ software (MassChroQ 2.2.21, PAPPSO, Jouy-en-Josas, France) [56] before data treatment and statistical analysis under R software (R 3.2.2, The R Foundation for Statistical Computing, Vienna, Austria). A specific R package called ‘MassChroqR’ was used to automatically filter dubious peptides for which the standard deviation of their retention time was longer than 30 s and to regroup peptide quantification data into proteins. For XIC-based quantification, normalization was performed to take account of possible global quantitative variations between LC-MS runs. The peptides that were shared between the different proteins were excluded automatically from the dataset as well as peptides that were present in fewer than 85% of the samples. Missing data were then imputed from a linear regression based on other peptide intensities for the same protein [57]. Analysis of variance was used to determine proteins with significantly different abundances between our two culture conditions.

#### 2.4.3. Proteomic Analysis

Subcellular location and lipoproteins were predicted using SurfG+, PSORTb, and PRED-LIPO [58,59,60]. The eggNOG-mapper v2 web tool was used to retrieve clusters of orthologous groups (COG) and KEGG protein categories [61]. Venn diagrams were obtained using Draw Venn Diagram [62] and volcano plots were conceived using VolcaNoseR [63]. Functional enrichment analyses were performed with the g: Profiler web server with the g: SCS multiple testing correction methods [64]. An input list of WC and EV proteins was compared to a custom-made Gene Matrix Transposed (GMT) file representing the theoretical proteome of *S. aureus* HG003 to identify statistically significant enriched COG and KEGG pathway terms (significance threshold of 0.05).

## 3. Results

### 3.1. Proteome of S. aureus HG003 and Its Derived EVs

Mass spectrometry analyses were performed to investigate the impact of different growth conditions on the proteome of the laboratory *S. aureus* HG003 strain and its derived EVs. WC and EV samples were recovered from early- and late-stationary growth phases (6 and 12 h, respectively) in the absence (V−) and presence (V+) of a sub-inhibitory concentration of vancomycin. Note that these same bacterial cultures were previously employed to characterize the RNA profile of HG003 and its derived EVs [44]. As previously reported by Luz et al., the EV preparation showed typical particle shape and size when analyzed by TEM and NTA (Figure 1A,B) [44]. The SDS-PAGE approach revealed a homogeneous protein profile between conditions in each group (Figure 1C); however, the EV protein profile was specific when compared to WC (Appendix A).

### 3.2. Protein Composition and Functional Characterization of S. aureus HG003 and Its EVs

Nano LC-ESI-MS/MS analyses identified 1111 unique proteins in all conditions, of which 967 were found in WC samples, and 556 were found in EVs samples (Appendix A). Principal component analysis (PCA) was first performed on peptide quantification values to assess the consistency of the proteomics data (Appendix A). The PCA scatter plot showed that samples from WC and EVs formed separate clusters. All the WC samples formed a unique cluster, while the EV samples were grouped into two clusters depending on whether the EVs were produced at 6 or 12 h. The impact of vancomycin availability on EV protein composition seemed to be limited compared to the incubation time. The PCA confirmed the consistency between biological replicates under each condition and pointed out the high impact of growth conditions on EV protein composition compared to WC.

Then, analyses were performed to investigate the composition and functions that were associated with WC and EV proteomes. The predictions of subcellular localization revealed that cytoplasmic proteins composed from 71 to 84% of the WC proteome, while these values dropped down to 42–55% for EVs (PSORTb, *n* = 668 vs. 236; SurfG+, *n* = 808 vs. 287; PRED-LIPO, *n* = 814 vs. 309) (Figure 2A–C). In contrast, WC presented less membrane proteins (6–10%) when compared to EVs (23–39%) (PSORTb, *n* = 133 vs. 220; SurfG+, *n* = 58 vs. 128; PRED-LIPO, *n* = 99 vs. 177) (Figure 2A–C). Interestingly, EVs seemed enriched with surface-exposed proteins (PSE) (SurfG+, *n* = 108 vs. 67) and lipoproteins (SurfG+, *n* = 44 vs. 23; PRED-LIPO, *n* = 41 vs. 22) (Figure 2B,C). Remarkably, proteins that were identified only in EVs (*n* = 144) corresponded mainly to membrane (*n* = 75) and PSE (*n* = 45) proteins, according to SurfG+ analysis.

Cellular processes that were related to COG also differed between WC and EV proteomes (Figure 2D). Generally, fewer EV proteins belonged to COG categories that were associated with “information, storage, and processing” when compared to WC (13.6% vs. 23.2%). On the other hand, COG categories that were related to “cellular processes and signaling” and “metabolism” accounted for more proteins in EVs (23.6% and 42.2%) compared to WC samples (17.8% and 40.8%). Remarkably, some COGs were twice more represented in EVs when compared to the WC, such as intracellular trafficking, secretion, and vesicular transport (U, 2.1% vs. 1.1%), defense mechanisms (V, 2.4% vs. 1.1%), and inorganic ion transport and metabolism (P, 12.5% vs. 5.7%) (Figure 2D). Note that similar results from the composition and functions of WC and EV proteomes were obtained when each growth condition was analyzed individually (Appendix A).

Functional enrichment analysis relative to the theoretical proteome of *S. aureus* HG003 also identified different enriched COG and KEGG categories for WC and EVs (adjusted *p*-value < 0.05). While both shared proteins that were related to metabolism, translation, and energy production, EVs were exclusively enriched with proteins that were linked to cell wall biogenesis, two-component systems, and transport systems (Figure 3). In summary, the vesicular proteome differs from that of WC, showing less cytoplasmic and more surface-exposed-, membrane-, and lipoproteins, being ~25% exclusive to EVs. Additionally, COG/KEGG enrichment analyses reinforce that functional protein profiles that are found in EVs significantly differ from those of the WC.

### 3.3. Cellular and EV Protein Composition Varies with Growth Conditions

The WC and EV protein compositions were compared between early- and late-stationary phases, in the absence or presence of vancomycin. From 1111 unique proteins identified with Nano LC-ESI-MS/MS experiments, 555 and 144 were exclusive to WC and EVs, respectively (Figure 4A, Appendix A). Proteins that were identified only in EVs included the enzymes thermonuclease (Nuc) and lipase (Lip), the adhesin Eap, and the Hld toxin. Although the different environmental conditions that were tested in this study resulted in several variations in protein content, the EV core proteome was represented by 340 proteins (61.1%), while this portion was higher for the WC core proteome (78.8%, *n* = 762) (Figure 4B,C). Proteins that were shared by EVs in all conditions comprised of virulence factors (Atl, Sbi, EbpS, Lip), metabolic enzymes (pyruvate dehydrogenase complex PdhABCD, Enolase), survival elements (SirA, FhuD1, FtnA), and resistance proteins (VraS, FmtA, PBPs).

Similar to a recent report that was published by our group on the RNA content of HG003 EVs in the same conditions, EV protein loads were higher at 6 h than at 12 h (515 vs. 433 proteins identified) [44]. EV proteins that were found at 6 h (*n* = 123), but not at 12 h, included autolysins (Sle1 and AcmB), the immunogenic protein SsaA, and the toxin LukH. Remarkably, proteins that were exclusive to EV samples at 12 h (*n* = 41) included Hld and PSMβ1. On the other hand, WC presented more proteins at 12 h than at 6 h (936 vs. 873). The analysis identified Pbp4, SepF, and GroES among those that were exclusive to WC at 6 h (*n* = 31), while RsfS, linked to ribosomal activity, the transcription regulator FapR, and the virulence factors Emp, SCIN, and LukG were present only in late-stationary growth phase samples (*n* = 94). Our data indicated a moderate antibiotic effect in both groups (EVs = 508 vs. 516; WC = 934 vs. 925, for the presence and absence of vancomycin, respectively). In EVs, 40 proteins were identified only in the presence of vancomycin (e.g., IsaB, FhuD1), with only 3 proteins that were found at both 6V+ and 12V+ conditions: the ribosomal protein RplQ, the cation/H+ antiporter MnhG, and the putative PitA protein that were linked to inorganic phosphate transport. Among the 48 EV proteins that were found only in the absence of vancomycin, 24 belonged to metabolic-related COG groups. In WC, the division protein DivlC belonged to the group of 33 proteins that were found only in the absence of vancomycin, while proteins that were related to DNA repair such as Nfo and Fpg were among the 42 proteins that were found only in the presence of vancomycin.

### 3.4. Cellular and EV Protein Abundance Varies with Growth Conditions

Quantitative analysis provided interesting information on the state of the cell and its derived EVs in different growth conditions (Appendix A). Regarding to the growth phase, 64 and 110 proteins were differently abundant in WC samples in the absence and presence of vancomycin, respectively. In WC samples without vancomycin, 25 proteins were more abundant at 6 h (e.g., Atl, IsaA, VicK), while 39 were more abundant at the late-stationary growth phase (e.g., IsaB, ModA, MreC) (Figure 5A). In the presence of vancomycin, 57 WC proteins were more abundant at 6 h (e.g., IsaA, PrkC, DltD), while 53 were more abundant at 12 h (e.g., Asp23, CcpA, IsaB, MreC) (Figure 5B). Compared to the WC samples, the protein content of EVs were less impacted by the growth phase, with only 23 and 9 proteins differently abundant in the absence and presence of vancomycin, respectively (Figure 5C,D). In both conditions, the virulence factor regulator SarR was more abundant at 6 h, and proteins AtpG, GlpF, TreP, and YqeZ were more abundant at 12 h (Figure 5C,D). In the absence of vancomycin, SitA, TagF1, and PTS transporters were among the 16 proteins that were richer at 12 h, while FusA was less abundant (Figure 5C). The presence of vancomycin hardly affected the protein abundance in either WC or EVs, with only a few exceptions. In WC, HemB and the lactonase Drp35 were less abundant at 6V− when compared to 6V+, presenting log2 fold change (log2FC) values of −2.14 and −5.01, respectively (Appendix A). In EVs, proteins that were more abundant in the presence of vancomycin at 12 h were the polypeptide cell division protein DivIVA (log2FC = 3.36) and the S-ribosylhomocysteine lyase LuxS (log2FC = 3.20) (Appendix A).

Astonishing differences were mainly observed in the relative protein abundance between WC and its derived EVs. Table 1 displays the top ten more and less abundant proteins in EVs relative to WC in different conditions. Ribosomal components and other proteins that were related to translation and transduction were generally less abundant in EVs when compared to WC (Appendix A). Conversely, key virulence factors that were related to adhesion and colonization, resistance, and survival were more abundant in *S. aureus* HG003-derived EVs than in the bacteria. Some proteins were more abundant in EVs whatever the condition that was tested (e.g., Atl, CamS, CydB, DltD, FatB, FecB, FtsH, HtrA, LtaS, ModA, SitA, YidC) (Figure 6, Appendix A). Other virulence factors such as the adhesin EbpS, the protein kinase PkrC, the multidrug resistance efflux pump EmrA, ABC transporters (e.g., AdcA, FhuD), the quinol oxidase complex QoxAB, and PBPs were more abundant in EVs in multiple conditions (Figure 6, Appendix A).

Some differences that were observed in the relative protein abundance between EVs and WC seemed to be affected by growth conditions. For instance, the immunoglobulin-binding protein Sbi, the secretory antigen SsaA, and the transcriptional regulator SarR were relatively more abundant in EVs than WC only at 6 h, while no significant differences were observed at 12 h (Figure 6). On the other hand, the histidine kinase VicK and the protein SecA were relatively enriched in EVs only at 12 h. Some differences were also observed regarding to the presence of vancomycin. PBP4 was relatively more abundant in EVs than WC at 6V+ vs. 6V− (log2FC = 3.78), while PBPX was more abundant in EVs compared to WC at V+ vs. V− in both growth conditions (Log2FC at 6 h = 2.96 and Log2FC at 12 h = 3.87). The moonlighting protein enolase was less abundant in EVs only in the presence of vancomycin (6V+ and 12V+), while EF-Tu and GAPA1 were less abundant in EVs than in the WC at 12 h (V− and V+). Our data showed that the protein content and abundance of *S. aureus* HG003 and its derived EVs vary according to the growth conditions. Moreover, important virulence factors were differentially abundant in WC and EVs, suggesting the selective packing of proteins into HG003 EVs.

### 3.5. Comparison between Protein and RNA Content of HG003 EVs

Comparative analyses between previously published HG003 EV RNome and proteomic data showed that of a total of 220 mRNAs [50], 116 (52.7%) were not found in the corresponding protein form in HG003 EVs. Another 88 mRNAs (40%) were present in HG003 EVs in at least one of the four conditions in their corresponding protein form. Finally, 16 elements (7.3%) were present in all conditions that were tested in both RNA and protein forms, including Eno, DnaK, Atl, QoXB, SecY, FusA, and PdhABC (Table 2).

## 4. Discussion

Since the first observation describing the EV release by *S. aureus* in 2009 [8], various studies have reported the EV production and secretion by several human clinical isolates and by strains that were isolated from other mammalian species [44]. The study of EVs biogenesis, release, and cargo sorting provides insight into the importance of these particles for bacterial physiology and survival, in addition to a better understanding of how EVs exert their functions [19,65,66,67]. However, to date, few *S. aureus* EV studies have considered the impact of different environmental conditions on vesicle production and composition [27,39,40,43,68]. A previous report by our group provided the first extensive characterization of the *S. aureus* HG003 EV RNome under four growth conditions: early- and late-stationary growth phases, and in the absence and presence of a sub-inhibitory concentration of vancomycin (0.5 µg/mL) [44]. Here, we used a proteomic approach to investigate how these conditions impact HG003 EV protein content. We also provide the first comparison of protein composition and abundance between EVs and their producing bacterial cells.

Our data indicated that HG003 EV proteome composition differed from that of the producing cells, presenting a higher proportion of membrane, surface-exposed and lipoproteins, and less cytoplasmic proteins compared to bacteria. Nevertheless, cytoplasmic proteins composed almost 50% of the EV proteome, an interesting feature that allows the secretion of proteins lacking export signals. The proportions that were observed here have already been reported in a study comparing EVs that were derived from five different *S. aureus* clinical and animal isolates, which presented similar protein compositions [25]. In addition to the different proportions that were observed in protein subcellular localization, HG003 EVs also presented exclusive features. We found that one-quarter of the total EV proteins were not detected in the bacterial proteome, although they were analyzed under the same conditions. It has already been suggested that Gram-negative bacteria manage to ‘exclude’ from EVs proteins that are abundant in the outer membrane (OM), which are usually associated with outer membrane vesicles (OMVs) biogenesis and composition [65,69,70]. In fact, EVs that are derived from various Gram-positive bacterial species are especially loaded with virulence factors [71,72,73,74]. Here, we demonstrate that most HG003 EV exclusive elements are membrane-associated or surface-exposed proteins, including the adhesin Eap and the δ-hemosylin toxin, which are key staphylococcal virulence factors. These differences in protein identification suggest differential cargo yield in EVs and bacteria, which may help EVs exert specific functions during infection.

Variations in the protein content of both WC and EVs were detected in different conditions. Regarding the growth phase, proteins that are exclusive to WC at 6 h included SepF, linked to cell division and GroES, a co-chaperonin that is essential in assisting protein folding [75,76]. These proteins are related to active dividing cells, corresponding to a bacterial population in the exponential growth phase. In contrast, the presence of the RsfS ribosomal silencing protein [77] and the FapR repressor of membrane lipid biosynthesis [78] at 12 h indicates *S. aureus* arrested cell division. In EVs, we found that immunogenic proteins such as SsaA are present only at the early-stationary growth phase. It has already been demonstrated that pre-inflamed lungs allow bacteria to hijack and resist host defense mechanisms in allergic asthma [79,80]. In this line, the presence of immunostimulatory molecules in EVs at lower cell densities could help to activate host cells even before bacterial arrival at distant sites, favoring *S. aureus* survival and persistence. It is also remarkable to note that LukH is present in EVs only at 6 h and in WC at all conditions, while LukG is found only in WC at 12 h. It has been demonstrated that individual components of LukH and LukG cannot promote host cell lysis, yet they each induce high levels of IL-8 release [81]. EVs may serve as vehicles to induce the host immune response at early infection states, while functional pore-forming toxin LukGH appears only later. Accordingly, Hld and PSMβ1 are exclusive to EVs at 12 h, which corresponds to the known toxin production by bacteria at stationary phases [6,82]. These pore-forming toxins promote host cell death and immunomodulation [83,84], a role that could be complementary played by EVs.

Regarding to the impact of antibiotics on the protein cargo, we observed that Nfo and Fpg proteins that were linked to DNA repair were identified in WC only in the presence of vancomycin. Accordingly, it has already been demonstrated that SOS response in *S. aureus* is slightly upregulated when facing cell wall inhibitory antibiotics [85,86]. Curiously, the putative inorganic phosphate PitA transporter that is found in EVs only in the presence of vancomycin has been previously associated with *S. aureus* tolerance to several antibiotics, including vancomycin [87,88].

Globally, our data showed that the conditions that were tested here had a more pronounced impact on the EV protein composition than on the WC. These results show that EV cargo is not an exact reflection of the protein cargo of producing cells. Slight variations applied to EVs may serve as a fine-tuning mechanism to help the bacterial population rapidly and efficiently respond to external conditions. Yet, despite the variations that were found, the EV core proteome still carries proteins that are linked to virulence (e.g., Ebps, Sbi), metabolism (e.g., Eno, PdhABCD), survival (e.g., FhuD), and resistance (FmtA, PBPs). Many of these elements are also found across several *S. aureus* strains [23,25] and may play active roles in the infection and pathogenesis of this bacterium.

Interestingly, differences in protein abundance were observed when comparing growth phases in WC and EVs. In WC, the Asp23 protein was more abundant at 12 h V+ compared to 6 h V+, but not in the absence of antibiotics. Accordingly, a 10-fold expression increase of the *asp23* gene has been shown after treatment with vancomycin [89]. The fact that Asp23 is more abundant at 12 h may reflect the accumulation of this protein after exposure to this antibiotic. In EVs, PTS transporters were more abundant at the late-stationary growth phase. Considering that bacteria face nutriment stress at high cell densities, EVs could help bacteria to improve sugar uptake. Finally, the impact of the absence and presence of antibiotics on protein abundance in the same conditions (either 6 h or 12 h) was scarce. In WC, the protein Drp35 was more abundant at 6 h in the presence of vancomycin. Accordingly, this protein acts in cells with perturbed membrane integrity, and its activation has been consistently reported in response to certain antibiotics, including vancomycin [90]. Moreover, it was interesting to note that LuxS was more abundant in EVs at 12 h in the presence of vancomycin. LuxS is involved in the synthesis of autoinducer 2 (AI-2), an essential element of the *S. aureus* agr quorum-sensing (QS) system [91]. Dysfunctional agr has been associated with glycopeptide intermediate-level resistant *S. aureus* (GISA) [92,93], and the use of QS inhibitors increases *S. aureus* susceptibility to vancomycin [94,95]. In addition, the QS also contributes to biofilm structuring and detachment through the activation and expression of proteases and other molecules with surfactant-like properties, such as PSMs [82,96,97,98]. In light of this, EVs that are derived from the stationary growth phase containing exclusive elements such as PSMβ1 may play a role in biofilm formation. It has already been shown that EVs that are purified from *S. aureus* BWMR22 strain that are grown in the presence of a sub-inhibitory concentration of vancomycin are able to increase bacterial adhesion and cell aggregation, contributing to *S. aureus* biofilm formation [99]. Interestingly, *S. aureus* EVs’ ability to increase surface hydrophilicity confers a competitive advantage by reducing biofilm formation by several other pathogenic bacteria, including *Acinetobacter baumannii*, *Enterococcus faecium*, and *Klebsiella pneumonia* [100]. Together, our data suggests that EVs may play a role in response to vancomycin stimuli and in biofilm formation through QS. The role of EVs in the physiology and fitness of the *S. aureus* global population may be addressed in future studies.

To our surprise, the most remarkable differences that were observed in this study were related to the relative protein abundance in EVs and bacteria. Even though ribosomes are highly abundant in cells [101], we found that proteins that are related to translation were relatively less abundant in EVs when compared to bacteria, while key virulence factors were more abundant. Highly loaded proteins in EVs included those that are involved in adhesion, colonization, survival, resistance, and modulation of the host immune response. One example is the Sbi protein, which was relatively more abundant in EVs at 6 h. This immunoglobulin-binding protein could interfere with complement activation, opsonophagocytosis and neutrophil killing [102,103]. In fact, it has been proposed that surface-localized Sbi protein in *S. aureus* ATCC14458 EVs may favor binding to host cells [8]. Interestingly, PBPX and PBP4 were relatively more abundant in EVs compared to bacteria only in the presence of vancomycin. We could speculate that the information that is perceived by bacteria in response to slight stimuli may somehow be transferred into EVs without largely affecting bacterial composition. These modified EVs could act as vehicles to promote bacterial resistance to antibiotics, among other important functions. Indeed, recent studies demonstrated that antibiotic stress contributes to the increased loading of β-lactam degrading proteins in *S. aureus* EVs, which protect ampicillin-susceptible bacteria in a dose-dependent manner [24,43].

Finally, although similar tendencies were observed between the HG003 EV protein and RNA content [44], such as EVs being more loaded at 6 h than 12 h, matches in coding mRNAs and proteins were not obvious. More than half of HG003 EV mRNAs [44] were not found in their corresponding protein forms. Some exceptions include the autolysin Atl, the moonlighting protein Eno, the elongation factor FusA, and the chaperone DnaK, which are present in all conditions in both RNA and protein forms. Their constant appearance in the EV proteome of several *S. aureus* strains and other species, including Gram-negative bacteria, suggests their role in EV biogenesis [25]. Indeed, Atl has already been associated with EVs release in *S. aureus* [41], while chaperones were shown to participate in selecting and packing proteins into EVs in Gram-negative bacteria [104,105].

## 5. Conclusions

This work provides a general proteomic picture of EVs that were derived from the *S. aureus* HG003 strain under different growth conditions. Additionally, comparing the EV proteome to that of producing cells in the same conditions brought new insights into *S. aureus* EV characteristics. Our data demonstrate that HG003 EV content and abundance are modulated by the environment and differ from that of bacterial cells, supporting the hypothesis that EV release is a mechanism that is ruled by selective cargo packing. New and exciting studies will continue to unveil the processes governing *S. aureus* EV biogenesis and production, and their role in bacterial physiology and pathogenesis.

## Figures and Tables

**Figure 1 microorganisms-10-01808-f001:**
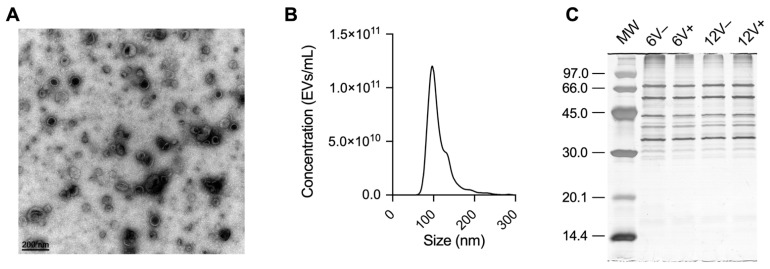
*S. aureus* HG003 EV characterization. (**A**) Cup-shaped EVs by transmission electron microscopy (TEM). (**B**) Monodisperse profile revealed by nanoparticle tracking analysis (NTA). (**C**) Protein profile of EV samples resolved in 12% SDS-PAGE. Molecular weight (MW) standards are indicated in kDa. Early- and late-stationary growth phases (6 and 12 h, respectively) in the absence (V−) or presence (V+) of vancomycin.

**Figure 2 microorganisms-10-01808-f002:**
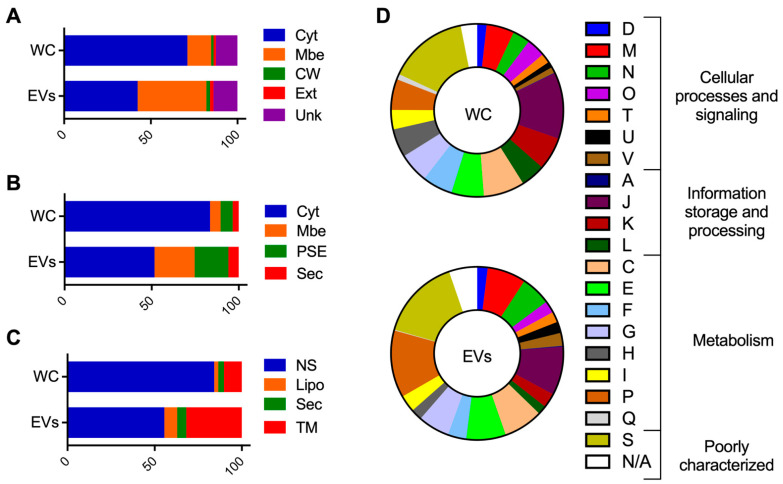
Comparative proteome analysis of *S. aureus* HG003 WC and EVs. The prediction of subcellular locations of proteins with PSORTb (**A**), SurfG+ (**B**), and PRED-LIPO (**C**), and the prediction of COG functional categories of proteins with eggNOG-mapper (**D**). Cyt, cytoplasmatic; Mbe, Membrane; CW, cell wall; Ext, extracellular; Unk, unknown; PSE, surface-exposed; Sec, secreted; NS, no signals found; Lipo, lipoprotein; TM, transmembrane; COG functional categories: D, Cell cycle control, cell division, chromosome partitioning; M, cell wall/membrane/envelope biogenesis; N, cell motility; O, post-translational modification, protein turnover and chaperones; T, signal transduction mechanisms; U, intracellular trafficking, secretion and vesicular transport; V, defense mechanisms; A, RNA processing and modification; J, translation, ribosomal structure, and biogenesis; K, transcription; L, replication, recombination, and repair; C, energy production, and conversion; E, amino acid transport and metabolism; F, nucleotide transport and metabolism; G, carbohydrate transport and metabolism; H, coenzyme transport and metabolism; I, lipid transport and metabolism; P, inorganic ion transport and metabolism; Q, secondary metabolites biosynthesis, transport, and catabolism; S, function unknown; NA, not available.

**Figure 3 microorganisms-10-01808-f003:**
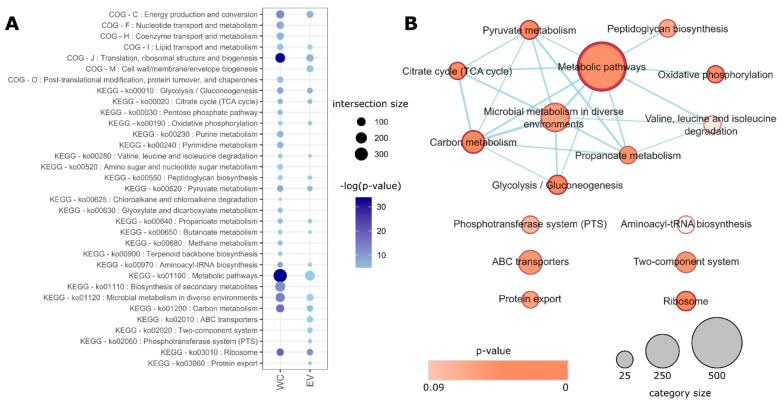
KEGG and COG categories that are enriched in *S. aureus* HG003 WC and EV proteomes. (**A**) Significantly enriched COG and KEGG terms that were associated with WC and/or EV groups. Darker colors represent more significant *p*-values (in logarithm scale) and bubble sizes represent the number of proteins in that category that were identified in the respective sample (the intersection size). (**B**) Enrichment map for EV-related KEGG terms, showing significant functional categories, their total sizes, and relationships (protein sharing).

**Figure 4 microorganisms-10-01808-f004:**
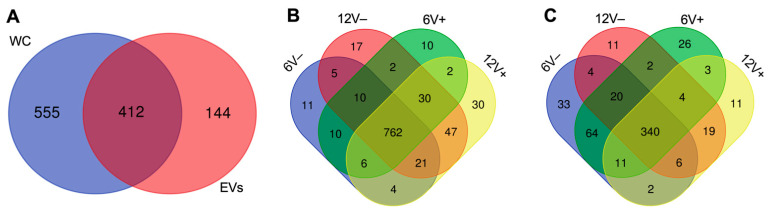
*S. aureus* HG003 WC and EV proteomes are differently modulated by growth conditions. Venn diagrams of all proteins that were identified in HG003 WC and EVs (**A**), and proteins that were identified in different growth conditions in WC (**B**) and EVs (**C**). Early- and late-stationary growth phases (6 and 12 h, respectively) in absence (V−) or presence (V+) of vancomycin.

**Figure 5 microorganisms-10-01808-f005:**
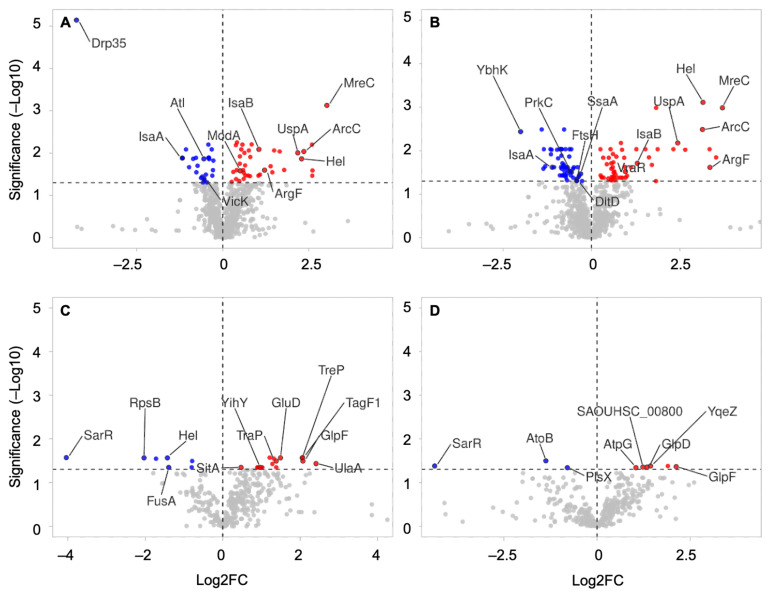
Volcano plot showing differently abundant proteins between growth phases. The graphs provide the log2 fold change (log2FC) ratios between 12 h and 6 h (12/6) conditions from WC without vancomycin (**A**), WC with vancomycin (**B**), EVs without vancomycin (**C**), and EVs with vancomycin (**D**). The vertical and horizontal dotted lines show the FC and threshold of significance (adjusted *p*-value < 0.05), respectively. The results are plotted in dots on a logarithmic scale, with negative log2FC in blue, positive log2FC in red, and nonsignificant values in grey.

**Figure 6 microorganisms-10-01808-f006:**
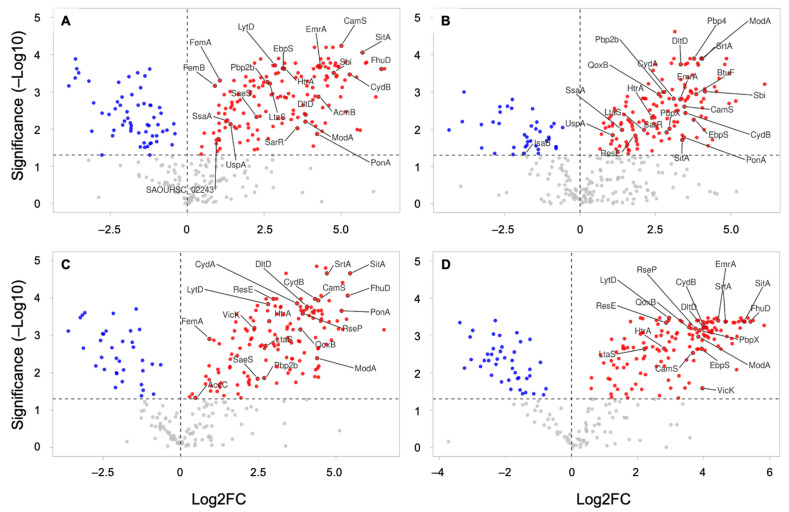
Volcano plots showing the relative protein abundance between *S. aureus* WC and its derived EVs. The graphs provide protein log2 fold change (log2FC) ratios between EVs and WC (EV/WC) conditions at 6 h without vancomycin (6V−) (**A**), 6 h with vancomycin (6V+) (**B**), 12 h without vancomycin (12V−) (**C**), and 12 h with vancomycin (12V+) (**D**). The vertical and horizontal dotted lines show the log2FC and significance threshold (adjusted *p*-value < 0.05), respectively. The results are plotted on a logarithmic scale, with negative log2FC in blue, positive log2FC in red, and nonsignificant values in grey.

**Table 1 microorganisms-10-01808-t001:** The top ten most abundant and less abundant proteins in EVs compared to WC in all conditions that were tested. Ratios correspond to the relative abundance of proteins (log2FC) between EVs and WC (EV/WC) conditions at 6 h without vancomycin (6V−), 6 h with vancomycin (6V+), 12 h without vancomycin (12V−), and 12 h with vancomycin (12V+).

6V−		6V+		12V−		12V+	
Protein	Ratio	Protein	Ratio	Protein	Ratio	Protein	Ratio
RplV	−3.08	Hup	−4.35	RpsJ	−3.62	Ctc	−3.38
RpsJ	−2.91	RplK	−3.91	RplM	−3.23	RplQ	−3.24
RpsI	−2.91	RplQ	−3.85	RpsE	−3.17	RpsE	−3.07
TraP	−2.87	Ldh	−3.27	RplR	−3.13	Ldh	−3.04
RplO	−2.82	PyrH	−2.88	RplL	−3.09	AtoB	−2.99
RpsE	−2.56	Ddh	−2.68	RplS	−2.58	RplM	−2.54
Ald	−2.43	RplL	−2.66	RpsK	−2.51	Pgk	−2.37
RplS	−1.78	RpsQ	−2.65	RpsD	−2.48	RpsD	−2.36
GluD	−1.52	RpsI	−2.41	Ddh	−2.4	Ddh	−2.31
RpsC	−1.29	RplO	−2.25	Pgk	−2.17	AdhP	−2.17
Atl	3.61	CyoA	3.14	Atl	4.38	* 00356	5.06
NupC	3.79	Pbp4	3.78	FatB	4.66	LpdA	5.1
CamS	4.01	ModA	4.02	SrtA	4.71	RecN	5.12
YidC	4.11	SrtA	4.05	SpsB	4.78	* 00717	5.16
* 00356	4.41	AgcS	4.61	PonA	5.18	SitA	5.22
SitA	4.56	PdhB	4.8	TcyA	5.21	SpsB	5.25
TcyA	4.64	TcyA	4.86	* 02587	5.34	TcyA	5.34
PdhB	4.67	LpdA	4.94	FhuD	5.38	FhuD	5.42
FhuD	5.05	* 00356	5.17	SitA	5.45	* 02587	5.5
LpdA	5.13	* 02650	6.13	PdhC	6.54	* 02650	5.83

* = SAOUHSC_ (locus tag). Example: SAOUHSC_00356.

**Table 2 microorganisms-10-01808-t002:** *S. aureus* HG003 elements that were found in EVs in both RNA and protein form in all the conditions that were tested.

Locus Tag	Product	Gene	SurfG+	COG
SAOUHSC_00994	bifunctional autolysin precursor, putative	*atl*	Sec	M
SAOUHSC_01683	molecular chaperone DnaK	*dnaK*	Cyt	O
SAOUHSC_00799	phosphopyruvate hydratase	*eno*	Cyt	G
SAOUHSC_00529	elongation factor G	*fusA*	Cyt	J
SAOUHSC_00206	L-lactate dehydrogenase	*ldh1*	Cyt	C
SAOUHSC_01040	pyruvate dehydrogenase α subunit, putative	*pdhA*	Cyt	C
SAOUHSC_01041	pyruvate dehydrogenase β subunit, putative	*pdhB*	Cyt	C
SAOUHSC_01042	branched-chain α-keto acid dehydrogenase	*pdhC*	Cyt	C
SAOUHSC_00796	phosphoglycerate kinase	*pgk*	Cyt	F
SAOUHSC_01002	quinol oxidase AA3, subunit II, putative	*qoxA*	PSE	C
SAOUHSC_01001	quinol oxidase, subunit I	*qoxB*	Memb	C
SAOUHSC_02478	50S ribosomal protein L13	*rplM*	Cyt	J
SAOUHSC_02506	30S ribosomal protein S3	*rpsC*	Cyt	J
SAOUHSC_00528	30S ribosomal protein S7	*rpsG*	Cyt	J
SAOUHSC_00527	30S ribosomal protein S12	*rpsL*	Cyt	J
SAOUHSC_02491	preprotein translocase subunit SecY	*secY1*	Memb	U

## Data Availability

The mass spectrometry proteomics data can be found at [106].

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
