# Peer review of "Impact of Environmental Conditions on the Protein Content of Staphylococcus aureus and Its Derived Extracellular Vesicles"

_microorganisms, 2022, doi:10.3390/microorganisms10091808_

Round 1

Reviewer 1 Report

   The manuscript (microorganisms-1860082) utilized label-free proteomics to investigate the impact of four growth conditions on the protein content of Staphylococcus aureus and its derived extracellular vesicles. The authors analyzed a large amount of protein data of S. aureus extracellular vesicles. However, the further validation analysis using PRM is necessary as such label-free analysis tends to bring large biases. Another problem to be aware of is that the analytical method used by the authors in the whole-cell analysis did not obtain a large amount of protein. Please find my comments below:

1.       Line 18, 199, 234, 272, 349: “S. aureus” should be changed as italic.

2.       Line 24: “WC” should provide the full name.

3.       Materials and Methods: The minimum inhibitory concentration (MIC) of the model strain HG003 against vancomycin should be added.

4.       Line 114: “14.000 rpm” should revised as “14,000 rpm”.

5.       Line 121-122: Please provide the manufacturer and model of SpeedVac concentrator.

6.       Line 123: “according to [55]…” please revise it.

7.       How to evaluate the quality of prepared extracellular vesicles of S. aureus?

8.       Line 147: “COGs” should provide the full name

9.       Line 160: “Whole-cell (WC)”, please delete whole-cell.

10.   Figure 1C: Why are so few low molecular weight proteins of extracellular vesicles? Too little sample amount of page glue? Can it be consistent with the LC-MS identification results?

11.   Line 178: “PCA” should provide the full name.

12.   Figure 2: “Cells” should be changed as “WC” in the figure.

13.   Line 212: “Clusters of Orthologous Groups (COG)”, please change as “COG”.

14.   Line 278: “Regarding the growth phase” changed as “Regarding to…”.

Reviewer 2 Report

I am not well-versed in proteomic, so my comments to this manuscript are of general nature and focus mainly on the introduction and discussion section. In my opinion, this is a very interesting and novel work. The methodology seems solid and the number of experimental replicates is sufficient. The choice of the four experimental conditions (6-12 h, +/- V) is obviously based on previous literature on the effect of sub-lethal antibiotic dose. Overall, I have only one comment/scientific curiosity: the EVs are knowingly involved in the preparation of the surface (host organism or inert) for bacterial colonization. Thus, it will great if the authors can add a short paragraph in the discussion section linking the EV production and proteome (with respect to the WC) to the early formation of biofilm under different environmental conditions. I think this will help attracting interest from the biofilm community and possibly providing ideas for longer-term experiments in th future.

Minor comments

Please italicize all the names of the microorganisms

Explain the acronym WC in the abstract

Author Response

Response to Reviewer 2 comments

Point 1: I am not well-versed in proteomic, so my comments to this manuscript are of general nature and focus mainly on the introduction and discussion section. In my opinion, this is a very interesting and novel work. The methodology seems solid and the number of experimental replicates is sufficient. The choice of the four experimental conditions (6-12 h, +/- V) is obviously based on previous literature on the effect of sub-lethal antibiotic dose. Overall, I have only one comment/scientific curiosity: the EVs are knowingly involved in the preparation of the surface (host organism or inert) for bacterial colonization. Thus, it will great if the authors can add a short paragraph in the discussion section linking the EV production and proteome (with respect to the WC) to the early formation of biofilm under different environmental conditions. I think this will help attracting interest from the biofilm community and possibly providing ideas for longer-term experiments in the future.

Response 1: The reviewer's comment is appreciated and we added a short paragraph of this matter in the discussion as suggested (lines 446-455 in the revised version of the MS).

Point 2: Please italicize all the names of the microorganisms

Response 2: Corrected as requested

Point 3: Explain the acronym WC in the abstract

Response 3: Mentioned as requested
